# Multiple-Attention Mechanism Network for Semantic Segmentation

**DOI:** 10.3390/s22124477

**Published:** 2022-06-13

**Authors:** Dongli Wang, Shengliang Xiang, Yan Zhou, Jinzhen Mu, Haibin Zhou, Richard Irampaye

**Affiliations:** 1School of Automation and Electronic Information, Xiangtan University, Xiangtan 411105, China; wangdl@xtu.edu.cn (D.W.); 201921002096@smail.xtu.edu.cn (S.X.); 2Shanghai Aerospace Control Technology Institute, Shanghai 201109, China; moujinzhen803@casa.cn; 3School of Mathematics and Computational Science, Xiangtan University, Xiangtan 411105, China; hb.zhou@smail.xtu.edu.cn (H.Z.); 202063910003@smail.xtu.edu.cn (R.I.)

**Keywords:** attention mechanism, adjacent position attention, cross-dimensional interactive, semantic segmentation

## Abstract

Contextual information and the dependencies between dimensions is vital in image semantic segmentation. In this paper, we propose a multiple-attention mechanism network (MANet) for semantic segmentation in a very effective and efficient way. Concretely, the contributions are as follows: (1) a novel dual-attention mechanism for capturing feature dependencies in spatial and channel dimensions, where the adjacent position attention captures the dependencies between pixels well; (2) a new cross-dimensional interactive attention feature fusion module, which strengthens the fusion of fine location structure information in low-level features and category semantic information in high-level features. We conduct extensive experiments on semantic segmentation benchmarks including PASCAL VOC 2012 and Cityscapes datasets. Our MANet achieves the mIoU scores of 75.5% and 72.8% on PASCAL VOC 2012 and Cityscapes datasets, respectively. The effectiveness of the network is higher than the previous popular semantic segmentation networks under the same conditions.

## 1. Introduction

Semantic segmentation occupies an irreplaceable position in the field of computer vision. Its main task is to label each pixel in the image with the correct semantic category label. Thanks to deep neural networks, semantic segmentation has been extensively researched and developed in recent years. With its rapid development, it has been applied to many fields, such as robot navigation, autonomous driving [1], medical image analysis, virtual reality, and agricultural model analysis [2,3]. Since the start of FCN [4], deep convolutional networks have been the main strategy. FCN is a pixel-level classification of images, that is, it classifies each pixel, thus solving the problem of semantic-level image segmentation. FCN can accept input images of any size and use the deconvolution layer to upsample the feature map of the last convolutional base layer to restore it to the same size as the input image, so that a prediction can be generated for each pixel while preserving the Spatial information in the original input image. Currently, the improved networks based on fully convolution networks (FCN) have achieved good results. However, due to the traditional model structure, only a small range of contextual information can be provided. The receptive field is small, and its limitation cause the segmentation effect to not reach the expected accuracy.

To solve the limitations issue of full convolution, Chen et al. proposed an atrous spatial pyramid pooling model in the Deeplab series in papers [5], which used multiscale dilated convolution to aggregate contextual information. Zhao et al. proposed a pyramid pooling model to capture contextual information [6]. However, the method of atrous convolution can only capture surrounding information and cannot be effective in global or dense context information. The method based on pyramid pooling cannot adaptively gather context information.

To aggregate the dense context information at the pixel level, nonlocal networks [7] use a self-attention mechanism to weigh the pixel features of each location with the pixel features of the whole image to obtain the long-distance dependence. Although this method achieves good results in visual tasks, it requires a huge attention map to calculate the relationship between each pixel pair. The complexity of $O(N^{2})$ leads to a long computation time and a large space occupation, where *N* is the size of the input feature map or the number of channels. Based on that idea, Liu et al. proposed a dual-attention network with complementary positional attention and channel attention [8], which generates a similarity matrix by calculating the relationship between each pixel in the feature map and the pixel in the whole image. However, not all pixels are correlated; we usually use matrix multiplication to calculate pixel similarity, and this calculation method obtains positive correlation weights. However, some goals in the segmentation of reality are not related or even opposed, such as cars and sky, sky and roads, etc. The method of calculating the similarity between each pixel and the pixels of the whole image space is not good for the segmentation of some objects, and the amount of calculation and storage complexity is large.

To solve the above problems, we propose a lightweight and effective positional attention module; our motivation is to replace the traditional single dense connected graph with two sparse connected graphs, unlike the existing network, which requires each pixel feature to be weighted with all pixels of the feature map. We only focus on the dependencies between pixels and neighboring pixels. Specifically, we firstly pay attention to the relationship between the pixels in the input feature map and the adjacent pixels in the same column to obtain a column attention map, and then assign the column attention weight to the input feature to obtain the column attention feature. Secondly, we calculate the similarity between the feature pixel and the adjacent pixels in the same row, and then assign the row attention weight to the column attention feature, thus forming a global positional attention. We named it the Adjacent Position Attention Module (APAM). This strategy greatly reduces the model complexity and computation time, and the parameter amount of the attention map is reduced from $O(N^{2})$ to $O(N\sqrt{N})$.

We compared the difference between the nonlocal block and the adjacent position attention block, as shown in Figure 1. When inputting the local feature map, the local feature obtains the attention map through matrix changes and operations, and then assigns the weight value in the attention map to the local feature V, so that each value of the local feature obtains a global weight, thereby obtaining a global context feature output. We improve the effect of the module by changing the calculation method of the attention map. The number of attention map weight values for each pixel in Figure 1a is (HW), and the attention weight value for each pixel in Figure 1b is Δ; the size of Δ is *H* or *W*, different from the dense connections adopted by the nonlocal module. As shown in Figure 1b, each position in the feature map is sparsely connected with other ones which are in the same column neighbors or the same row neighbors in our block. Taking the five pixels on the diagonal as an example, the first pixel (red) only pays attention to three adjacent pixels in the same column (the first column), the second pixel (yellow) only pays attention to the four adjacent pixels in this column, and the middle pixel (blue) pays attention to all the pixels in this column. The number of concerned pixels increases linearly from the edge to the middle. Leading to the predicted attention, the map only has about $O(2\sqrt{N})$ weights rather than O(N) in the nonlocal module.

In semantic segmentation, the low-level features have a higher resolution with more position and detail information; the high-level features have a large Receptive Field, so there is rich semantic and category information. We should combine the advantages of the two. The previous methods used jump structure for fusion [4], Chen [9] merged the features after the atrous spatial pyramid pooling with the low-level features; U-Net [10] and SegNet [11] use encoding and decoding structures for fusion. These fusion methods are too simple and the effect is not obvious. Later, DFN [12] and PAN [13] used the global average pooling of high-level features to guide the fusion of low-level features. Global average pooling is suitable for extracting the feature information of large target objects, and it is easy to confuse surrounding objects for small target objects. To balance the fusion of high and low-level features, we propose a cross-dimensional interactive attention mechanism to solve this problem. By capturing rich feature representations across dimensions, the cross-dimensional interactive attention mechanism consists of three branches. The first two branches are responsible for capturing the dependency between the channel dimension (C) of the input feature and the spatial dimension (*H* × *W*), and the latter branch is composed of the spatial attention mechanism. High-level feature maps and low-level feature maps are jointly input into the cross-dimensional interactive attention model, so that the detailed information of low-level features and the semantic information of high-level features are perfectly combined, so that the semantic segmentation of the image achieves a good effect.

In summary, our contributions are as follows:To capture the long-distance dependence, we proposed the APAM, which combined with the Channel Attention Module to form a new dual-attention model (NDAM), which is lighter and more effective than DANet [8].To obtain a better semantic segmentation effect, we designed a cross-dimensional interactive attention feature fusion module (CDIA-FFM) for fusing features from different stages in the decoder.Combining NDAM and CDIA-FFM, a new network MANet for semantic segmentation is proposed. It has obtained good results in the two benchmark tests PASCAL VOC 2012 and cityscapes.

The rest of the paper is organized as follows. We review related work in Section 2 and describe the APAM and CDIA-FFM in detail in Section 3, and then introduce the entire network framework. In Section 4, we present the fusion experiments, experiment comparisons, experimental results, and the analysis of the experimental results. Section 5 focuses on conclusions and future work suggestions.

## 2. Related Work

### 2.1. Semantic Segmentation

Semantic segmentation has been a hot research topic in recent years. FCN is the first semantic segmentation method that uses full convolution for end-to-end learning. It is called the groundbreaking work of semantic segmentation. Later, the improved network based on FCN also achieved good results. Uet [10], Deeplabv3+ [9], RefinNet [14], and DFN [12] adopt coding and decoding structures to fuse low-level and high-level features for dense prediction. To deal with objects of various scales and shapes, the adaptive scale convolution (SAC) and deformable convolution (DCN) methods have improved the standard convolution operator. AFF [15] uses adversarial learning to capture the semantic relationship of adjacent pixels in space. SANet [16] proposed pixelwise prediction and pixel grouping, which decompose semantic segmentation into two subtasks. The first subtask requires precise pixel-by-pixel labeling and introduces spatial constraints into image classification. The second subtask is pixel grouping. This task directly encourages the grouping of pixels belonging to the same category without space constraints. The final segmentation result is combined with the output of the four stages of the network to integrate multiscale context. OCRNet [17] characterizes pixels by using the representation of the corresponding object class. Firstly, learn the target area under the supervision of ground truth segmentation. Secondly, represent the object area by gathering the representation of the pixels in the object area. Finally, the relationship between each pixel and each target area is calculated, and the object context representation is used to enhance the representation of each pixel. Semantic segmentation has also been studied in domain adaptation and data reasoning, semisupervised [18], weakly supervised [19], and Few-Shot Learning [20].

### 2.2. Encoding and Decoding

The role of the encoder is to capture more advanced semantic information, and the size of the feature map gradually becomes smaller. The purpose of the decoder is to restore the details of the feature and the spatial scale. The encoder receives the input image, learns the feature map of the input image through the neural network, and the decoder gradually implements each pixel category labeling on the feature map. There are various encoding structures in the segmentation task, but the decoders are similar and also play a key role in the segmentation. Many computer vision tasks use encoding and decoding structures, such as medical image segmentation [21,22], Human Pose Estimation [23], Object Detection [24], etc.

### 2.3. Attention Module

The advantage of the attention module is that it can aggregate contextual information. Recently, researchers have proposed a variety of attention mechanism modules. For example, SENet [25] is the Channel Attention Module, which acquires the importance of each channel through learning. Adaptive recalibration of the characteristic response of the channel, CBAM [26], first achieves channel-level attention through the Channel Attention Module, and then through a spatial attention module to achieve spatial attention. DANet [8] proposed a novel dual-attention network, which uses the self-attention mechanism to improve the discriminative ability of feature representation in scene segmentation. CCNet [27] proposed a new cross-attention module, which can more effectively obtain the context dependency of the whole image. BAD [28] proposed a lightweight bilateral attention decoder for real-time semantic segmentation. Segmenter [29] proposed a new semantic segmentation method based on Vision Transformer, which uses multihead attention to model the global context.

Different from previous methods, our proposed spatial attention module is lightweight and effective. It replaces a single dense connected graph with two sparse connected graphs. We first focus on the relationship between a pixel and its upper and lower adjacent pixels, and then pay attention to the relationship between the pixel and its left and right neighboring pixels to form a full image of attention. Then, the spatial attention and channel attention are added in parallel to achieve good results. In addition, we carry out multistage feature fusion through the encoding and decoding structure. Unlike simple fusion methods, we combine the cross-dimensional interactive attention mechanism to capture the dependency between the different dimensions of the feature map, highlight the detailed information of low-level feature maps and the semantic information of high-level feature maps, highlight the feature representation of the fusion, and further improve the accuracy of semantic segmentation.

## 3. Method

In this section, we give the details of the proposed new dual-attention model (NDAM) and the feature fusion model (CDIA-FFM), and we describe the entire network MANet of the semantic segmentation task.

### 3.1. Network Architecture

As shown in Figure 2, we choose ResNet101 [30] pretrained on ImageNet as the backbone network, and use dilated convolution in the backbone network. We propose a new dual-attention model (NDAM) that is placed at the end of the base network to explore global contextual information by establishing associations between features through an attention mechanism. The cross-dimensional interactive attention feature fusion module (CDIA-FFM) is proposed as a decoder to enhance the fusion of high-level and low-level features, so that the semantic information and location information in the fusion module can be explicitly expressed. The backbone network, NDAM, and CDIA-FFM together form the overall framework MANet.

### 3.2. New Dual-Attention Model

#### 3.2.1. Overview of the New Dual-Attention Model

The new dual-attention model (NDAM) consists of the Adjacent Position Attention Module (APAM) and Channel Attention Module. As shown in Figure 3. The local feature map $F\in R^{C\times \;H\times W}$, where C,H,W represent the number of channels, space height, and width, respectively, *F* represents the local features of the input, Q,K,V are the matrix vectors generated according to the input features, *V* can be regarded as the matrix vector of the input features, and *Q* and *K* are the feature matrix for calculating the attention weight. *Q* and *K* calculate the similarity, pass this similarity value through the Softmax layer to obtain a set of weights, and obtain the output feature matrix according to the sum of the products of this set of weights and the corresponding *V*.

We propose the Adjacent Position Attention Module (APAM) to enhance the representative ability of pixels. As shown in Figure 3A, the Adjacent Position Attention Module applies two convolutional layers with 1 × 1 filters on *F* to generate two feature maps *Q* and *V*, respectively. The Adjacent Position Attention Module is completed in two steps: (1) The feature matrix $R^{C_{k}\times H\times W}$ and the matrix *K* are used for einsum operation to obtain the column attention map, and the operation result is used for the einsum operation with the feature matrix $R^{C_{v}\times H\times W}$. This process is equivalent to taking the column attention. The force weight is assigned to the input feature, so that the attention feature map is obtained, and then the normalization process is performed; (2) Perform the einsum operation on the feature matrix and feature matrix $Q_{1}$ to obtain the row attention map, and the weight of the row attention map is assigned to the features with column attention through the einsum operation, thereby obtaining a global dependency.

The high-level feature map of each channel represents the response of a specific class, and different channels are interrelated; establishing the interdependence between channels can improve semantic expression. The channel attention refers to the PAM in DANet [8]. As shown in Figure 3B, reshape and transpose the feature *F* to obtain the feature matrix $R^{C\times N}$, where *N* represents the product of high *H* and high *W* of the feature map, then reshape the *F* ture *F* to the matrix $R^{N\times C}$ and multiply these two matrices to obtain the channel attention map $D\in R^{C\times C}$, the matrix multiplication of the channel attention map *D* and reshaped *F*, and the result is reshaped into $R^{C\times H\times W}$. Then, the result is multiplied by the adaptive parameter α and then summed with the feature map *F*, element by element to obtain the output feature *E*. The output of the adjacent position attention mechanism is multiplied by the adaptive parameter β and then added element-wise with the output of the Channel Attention Module to obtain the final attention feature map $F^{\prime }$. Its definition is as follow:(1)$$F^{\prime }=\alpha C_{a}+\beta P_{a}+F$$
where parameters α and β are weights that can be gradually learned from 0, $C_{a}$ and $P_{a}$, respectively, represent the output of the channel attention model and the location attention model, and *F* represents the input local feature map. The adjacent position attention model and the channel attention model are added in parallel to obtain a new dual attention.

#### 3.2.2. Details of Adjacent Position Attention

The Adjacent Position Attention Module (APAM) consists of a column attention layer and a row attention layer. We analyze the column attention layer and the row attention layer from a two-dimensional perspective. Let $▵=\left\{-\frac{H}{2},\ldots ,0,\ldots ,\frac{H}{2}\right\}$ be a set of offsets of *H*, $K\in R^{▵\times C_{K}}$ represents the position embedding matrix, *H* denotes a variable learnable relative position embedding, and the *K* matrix consists of randomly initialized values and zero elements. The position of the zero element is variable and changes according to the position of interest. We only pay attention to the adjacent positions of the pixels, and the column attention layer only pays attention to the column adjacent positions of the pixels. The number of adjacent positions of interest increase linearly from *H*/2 to *H*, that is, the most edge pixel only pays attention to half of the column pixels of the feature map, and the middle position pixel pays attention to the entire column of pixels. *K* represents the key in self-attention. Let $V_{xy}\in R^{H\times C_{v}}$ be a matrix of *H* adjacent values in the same column of pixel $\left(x,y\right)$, and $f_{xy}$ represents the output of the pixel $\left(x,y\right)$ through the column of the interest layer.
(2)$$f_{xy}=\left(q_{xy}k^{T}\right)V_{xy}$$
where $q_{xy}$ represents the query of pixels. The row attention layer is similar to the column attention layer, and its position offset ${▵}_{1}=\left\{-\frac{W}{2},\ldots ,0,\ldots ,\frac{W}{2}\right\}$, *K* and *V* are adjusted accordingly, $K_{1}\in R^{triangle_{1}\times C_{k}}$, $V_{xy}^{\prime }\in R^{W\times C_{v}}$. The definition of the line attention layer is as follows:(3)$$f_{xy}^{\prime }=\left(q_{xy}k_{1}^{T}\right)V_{xy}^{\prime }$$
Because each pixel only pays attention to *H* adjacent pixels in the same column, or only pays attention to *W* adjacent pixels in the same row, the calculation amount and memory complexity of the row and column attention layers are O(NH) and O(NW), respectively.

The Adjacent Position Attention Module can capture contextual information in the horizontal and vertical directions. The feature map passes through the column attention layer and the row attention layer to form a global spatial attention module. As shown in Figure 4, all pixels from x to $x^{\prime }$ in (a) are calculated for the similarity of adjacent pixels in the same column, and the context information of *H* pixels in the same column is obtained. The feature map passes through the column attention layer (a), and the output result is after the line attention layer (b); all pixels from y to $y^{\prime }$ in (b) are calculated for the similarity of adjacent pixels in the same line, the W pixel context information of the same peer is obtained, so the pixel (x,y) in Figure 4 obtains the context information of the *H* × *W* area, that is, each pixel in the attention module of the adjacent position can collect the context information of the *H* × *W* area to generate new features with dense and rich contextual information.

### 3.3. Cross-Dimensional Interactive Attention Feature Fusion Module

#### 3.3.1. Cross-Dimensional Interactive Attention Mechanism

We study a lightweight and effective attention mechanism that is used to highlight features, a new method for extracting features by cross-dimensional interaction composed of three branch structures, and named it the cross-dimensional interactive attention mechanism. As shown in Figure 5a, for input feature maps, cross-dimensional interactive attention establishes the dependency between dimensions through rotation transformation and residual operation. The previous channel attention has its specific role. However, channel attention involves the loss of location information in pooling operations. CBAM [26] introduced spatial attention as a supplementary module for channel attention. To put it simply, spatial attention means “where in the channel to focus”, and channel attention means “which channel to focus on”. The disadvantage is that channel attention and spatial attention are independent and separate calculation processes, and there is no mutual dependence between the channel dimension and the spatial dimension. Inspired by the way of constructing spatial attention, we proposed the concept of cross-dimensional interaction; this shortcoming is solved by the interaction between the captured space dimension and the channel dimension. We introduce cross-dimensional interactions in the three branches to capture the dependencies between the (C,H), (C,W), and (H,W) dimensions of the input features. C-Pool, H-Pool, and W-Pool represent pooling operations in the channel, height, and width dimensions, respectively. This reduces the depth and retains the rich representation. It cats the maximum pooling feature and average pooling feature in *C*, *H*, and *W* dimensions, respectively. Taking C-Pool as an example, its expression is:(4)C-Pool=cat[MaxPool(I),AvgPool(I)]
where *I* is the input feature and its shape is (*C* × *H* × *W*). After C-Pool operation, the result is a feature tensor with shape (2 × *H* × *W*).

The cross-dimensional interactive attention mechanism consists of three branches. It receives an input feature map and outputs a refined feature map of the same shape, as shown in Figure 5a, given an input feature $I\in R^{C\times H\times W}$ is passed to three branches, respectively. In the first branch, the number of channels of input features is reduced to two by C-Pool, then goes through an *N* × *N* convolutional layer, batch normalization processing, and then generates the attention weight of shape 1 × *H* × *W* through the sigmoid activation layer, and finally assigns the weight to the input feature tensor; it extracts the information between the height and width dimensions, referred to as the spatial attention mechanism. In the second branch, the input feature is converted into $I_{1}\in R^{H\times C\times W}$ through the permute operation, and the dependency between the channel and the width is captured through the H-Pool, and then through the convolution layer, the normalization operation, and the activation layer. The obtained weights give the shape to the intermediate feature of *H* × *C* × *W*, and finally perform the permute operation to transform the shape of the feature back to *C* × *H* × *W*. The third branch uses the permute function to transpose the dimension of the input feature tensor to $I_{2}\in R^{W\times H\times C}$; after the W-Pool operation, the dependency between height and channel is captured. Then, through operations such as convolutional layers, the generated weights are assigned to the feature tensors. Finally, it passes through the dimensional transposition function. The cross-dimensional interactive attention mechanism can be represented by the following equation:(5)$$Y_{1}=I\sigma \left(\varphi \left(f^{N}\left(P\left(I\right)\right)\right)\right)$$
(6)$$Y_{2}=Pm\left(I_{1}\left(\varphi \left(f^{N}\left(P_{1}\left(I_{1}\right)\right)\right)\right)\right)$$
(7)$$Y_{3}=Pm\left(I_{2}\left(\varphi \left(f^{N}\left(P_{2}\left(I_{2}\right)\right)\right)\right)\right)$$
(8)$$Y=\frac{1}{3}\left(Y_{1}+Y_{2}+Y_{3}\right)$$
where *P*, $P_{1}$, and $P_{2}$ represent C-Pool, H-Pool, and W-Pool, respectively. $f^{N}$, φ, and σ denote convolution, batchnorm, and sigmoid operations, respectively. $Y_{1}$, $Y_{2}$, and $Y_{3}$ stand for branch 1, 2, 3, and the whole module, and the parameter $\frac{1}{3}$ is obtained after many experiments.

#### 3.3.2. Feature Fusion Module

As shown in Figure 5b, we embed the cross-dimensional interactive attention into the feature fusion module. The high-level feature A reduces the dimensionality through a 1 × 1 convolutional layer, and then increases the size of the feature map through upsampling to obtain feature map C of the same size and dimension as the low-dimensional feature map; feature map C and the low-level feature map B perform cat and 3 × 3 convolution operations to obtain feature map D, which enables the simple fusion of high-dimensional features and low-dimensional features, then performs the cross-dimensional interactive attention operation on the feature map D to highlight the information of the fusion feature map and obtains a feature map that integrates semantic information and location information. Cross-dimensional interactive attention can not only capture the dependencies between different dimensions but also highlight the semantic information in high-dimensional features and detailed information in low-dimensional features. The whole network has carried out a total of three high-level and low-level fusions. According to the size of the feature map, the size of the convolution *N* × *N* in the cross-dimensional interactive attention mechanism is different. The convolution sizes of the three fusions are 3 × 3, 5 × 5, and 7 × 7; convolution expands as the size of the feature map expands, which is conducive to expanding the receptive field.

## 4. Experiments

To verify our proposed method, we conducted evaluation experiments on two authoritative semantic segmentation datasets. They are the PASCAL VOC 2012 dataset [31] and the city scene dataset Cityscapes [32], and we have performed all ablation experiments on the PASCAL VOC 2012 dataset.

### 4.1. Datasets and Implementation Details

PASCAL VOC 2012: As one of the semantic segmentation benchmark data sets, it is often used in comparative experiments and network model effect evaluation. It contains 20 object categories and a background category. The training set contains 1464 images, the validation set contains 1449 images, and the test set contains 1456 images.

Cityscapes: The data set has a total of 5000 images, including street scenes in 50 different cities, 19 semantic class high-quality pixel-level labels, and a background. The pixels of each image are 2048 × 1024. A total of 2979 images in the data set are for training, 1525 images are for testing, and 500 images are for verification

There are three main criteria for measuring the accuracy of image semantic segmentation, which are pixel accuracy, average pixel accuracy, and average IOU. Our experiment uses the commonly used criterion mIOU, which calculates IOU for each category separately, and then averages the IOU of all categories. In semantic segmentation, there are two sets of real labels and predicted values; IOU is to calculate the ratio of intersection and union of these two sets.
(9)$$\mathrm{mIOU}=\frac{1}{K+1}\sum_{i=0}^{k}\frac{P_{ii}}{\sum_{j=0}^{k}P_{ij}+\sum_{j=0}^{k}P_{ji}-P_{ii}}$$
Assuming there are K+1 classes (including a background), $P_{ij}$ represents the number of pixels that belong to class *i* but are predicted to be class *j*. $P_{ii}$ represents the number of correct predictions, and $P_{ij}$ and $P_{ji}$, are interpreted as false positives and false negatives, respectively.

### 4.2. Implementation Details

Training settings: We build our network based on the PyTorch-1.6 experimental environment and train the network on a server with 2 NVIDIA GeForce GTX 1080Ti. Stochastic Gradient Descent (SGD) with mini-batch is used for training. We use a “poly” learning rate strategy, where the initial learning rate is multiplied by ${\left(1-\frac{iter}{max\_iter}\right)}^{power}$ with power = 0.9. We perform data augmentation by randomly scaling (from 0.5 to 2.0) the input images and flipping the method horizontally during training. For Cityscapes, the initial learning rate and weights are 0.01 and 0.0001, respectively, then high-resolution patches (768 × 768) from the resulting images are randomly cropped out. We set the training time to 60 epochs for cityscapes, and the batch size is 4. For PASCAL VOC 2012, the initial learning rate and weights are 0.007 and 0.0005, respectively, the crop size is 513 × 513, the batch size is 8, and the training time is set to 50 epochs.

### 4.3. Results on PASCAL VOC 2012

#### 4.3.1. New Dual-Attention Model

We set NDAM at the end of the base network resnet-101 to capture the spatial position dependence of features and the correlation between channels. It significantly improves the segmentation results by modeling rich context correlations on local features. To verify the effectiveness of NDAM, we conducted corresponding comparison and ablation experiments.

In the spatial correlation modeling of the acquired features, we conducted five sets of comparative experiments, as shown in Table 1. The effects achieved are different depending on the position of the attention mechanism. The first and second sets of experiments compare our proposed neighbor location attention with traditional spatial attention. In the third set of experiments, the location attention acquisition method is to calculate the similarity between all pixels and the pixels of the whole image. In the fourth set of experiments, the acquisition method is to calculate the similarity between the pixel and all the pixels adjacent to the same column and in the same row in turn. In the fifth set of experiments, the acquisition method is to calculate the similarity of pixels in sequence with the pixels adjacent to the same column and in the same row. After comparison, the experimental program of the fifth group has the best effect, and its mIOU is 73.57%.

In order to compare the parameter amount and computational complexity of NDAM and DANet, it can be obtained from Table 2. The parameter amount of NDAM is 1.033 M less than DANet, and the FLOPs of NDAM are 1.036 G smaller than DANet. Combining Table 1 and Table 2 under our experiment conditions, the effect of NDAM is 0.18% better than DANet, and the amount of parameters is smaller.

To understand our model more deeply, we visualized the effect of semantic segmentation, as shown in Figure 6. The first and second columns represent the original image and the label image, respectively. The third column and the fourth column, respectively, represent the DANet and NDAM segmentation results. In the segmentation result of the first image, our model has a more complete contour of the rider. The shape of the bicycle is more obvious in the segmentation result of the second image. In the segmentation result of the third image, our model has fewer red error segmentation pixels. This verifies the effectiveness of the cross-dimensional interactive attention fusion module, which makes the boundary and contour of the target object in the segmentation result more perfect. Overall, the NDAM segmentation results are better.

#### 4.3.2. Cross-Dimensional Interactive Attention Feature Fusion Module

The CDIA-FFM can not only capture the dependencies between dimensions but also highlight low-dimensional features and high-dimensional features. We conducted an ablation experiment on the setting of the convolution kernel size in the cross-dimensional interactive attention feature fusion module, as shown in Table 3. The three convolution kernels are, respectively, set in the three fusion modules. The network effect is the best when the sizes of the convolution kernel are set to 3, 5, and 7, respectively. The size of the convolution kernel is set according to the size of the feature map in the fusion module, and the convolution kernel increases with the size of the feature map.

To obtain a better segmentation effect, we combined the NDAM and CDIA-FFM to form MA-FFNet, as shown in Table 4. We compare MA-FFNet with some of the existing semantic segmentation networks, such as [8], DeepLabv3+ [9], OCRNet [17], EfficentFCN [33], and Bisenet v2 [34]. Our segmentation effect is 3.1% higher than that of the basic network, and the effect is higher than the comparison network under the same experimental conditions, so our network is feasible. We randomly visualized several segmentation results. As shown in Figure 7, it is obvious that the segmentation effect of our network is better. Comparing the chairs, bicycles, and sheep legs in the figure, all can prove the effect of our segmentation network.

### 4.4. Results on Cityscapes

To further verify the effectiveness of our network, we conducted verification and comparison experiments on a very challenging cityscapes data set. As shown in Table 5, we have achieved very good results. The Miou of 72.8% is much higher than the result of the comparison experiment and 6.19% higher than the base network.

We visualized the segmentation results on the cityscapes data set. As shown in Figure 8, the segmentation diagram of telephone poles and traffic signs has the best segmentation effect in our network, which illustrates the effectiveness of the Adjacent Position Attention Module, calculates the similarity between the pixel and the adjacent pixel in the column, so that this long target object receives attention. One disadvantage of our network is that it would segment the Mercedes-Benz car head (as example) into other targets in the image.

### 4.5. Compared with Transformer Method

With the development of computer vision, the visual transformer has been effectively used in semantic segmentation, and we compare our proposed segmentation network with the existing transformer methods. As shown in Table 6, the performance of our method is compared with Segformer [35]. There are five backbone networks with different parameters in Segformer, and Mit-B2 with similar parameters to our method is used as backbone. Under the same training epoch and different training strategies (Segformer uses the original training strategy in the comparison experiment), our method performs 0.6% higher than Segformer on the PASCAL VOC 2012 dataset and lower than Segformer on the cityscapes dataset. The main reason is that our network is not pretrained. If our method is effectively pretrained, its semantic segmentation performance is comparable to the transformer semantic segmentation network.

## 5. Conclusions

In this paper, we propose a MANet for semantic segmentation. Specifically, we designed the NDAM and CDIA-FFM. The NDAM improves the traditional nonlocal form of the position attention module, in which the adjacent position attention mechanism first calculates the similarity between the pixel and the adjacent pixels in the same column, and the result is the similarity between the pixel and the adjacent pixels in the same row to form a global dependency. The combination of the Adjacent Position Attention Module and the Channel Attention Module can effectively capture long-distance contextual information. The CDIA-FFM is a decoding module; during the feature downsampling process of the semantic segmentation network, some details such as contours and boundaries are lost. To obtain a better segmentation effect, the decoding module fuses the low-level feature map with the high-level feature map, and then cross-dimension is performed after fusion. Interactive attention processing makes the detailed information in the low-dimensional feature map and the semantic information in the high-dimensional feature map stand out, thereby obtaining a perfect fusion. Experiments show that the segmentation results are more accurate under the joint action of the two modules. Our network has achieved excellent results on both the PASCAL VOC 2012 and cityscapes datasets. In addition, the computational complexity of the decoding module of our network needs to be reduced; we will conduct research in the future. Full supervision requires a large amount of label data. In real projects, label data is limited. Using semisupervised or weakly supervised learning strategies seems promising for future research.

## Figures and Tables

**Figure 1 sensors-22-04477-f001:**
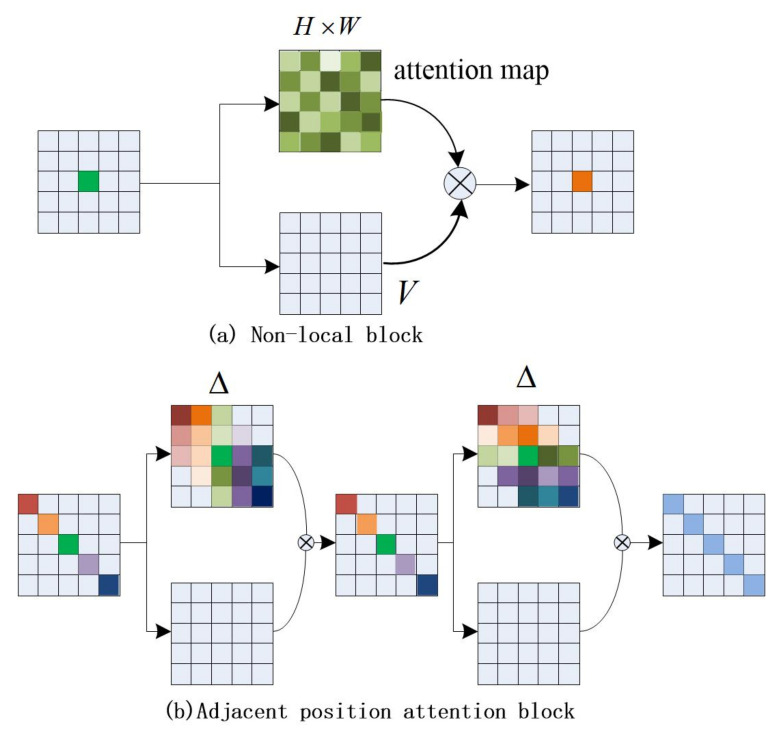
Diagram of two attention-based contextual aggregation methods.

**Figure 2 sensors-22-04477-f002:**
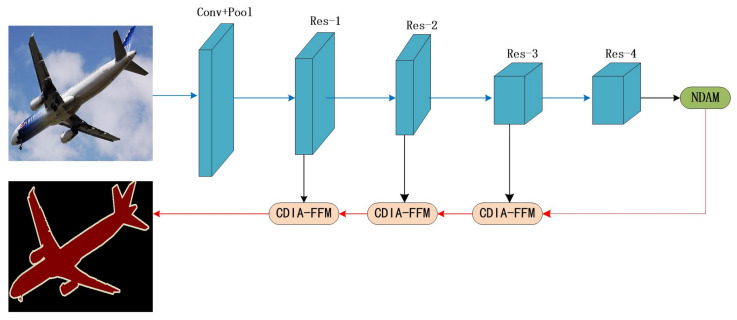
Overview of MANet for semantic segmentation. We use ResNet101 to extract dense features. We use NDAM to effectively capture long-distance contextual information and use CDIA-FFM to fuse features at different stages. The blue and red lines represent the downsample and upsample operators, respectively.

**Figure 3 sensors-22-04477-f003:**
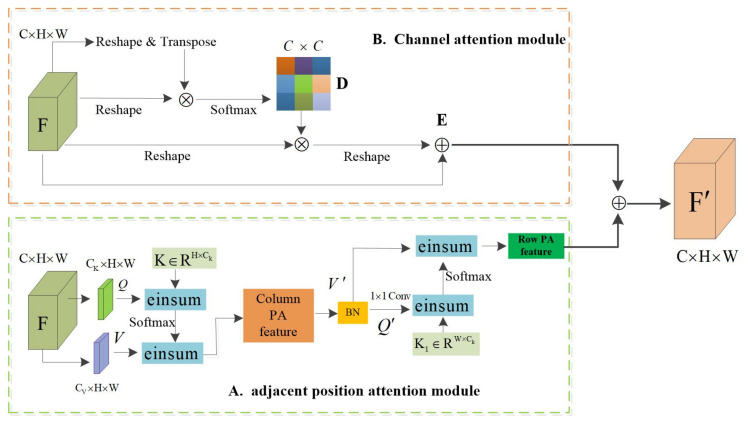
The overall framework of the new dual-attention model (NDAM). The details of the Adjacent Position Attention Module and Channel Attention Module are illustrated in (**A**,**B**).

**Figure 4 sensors-22-04477-f004:**
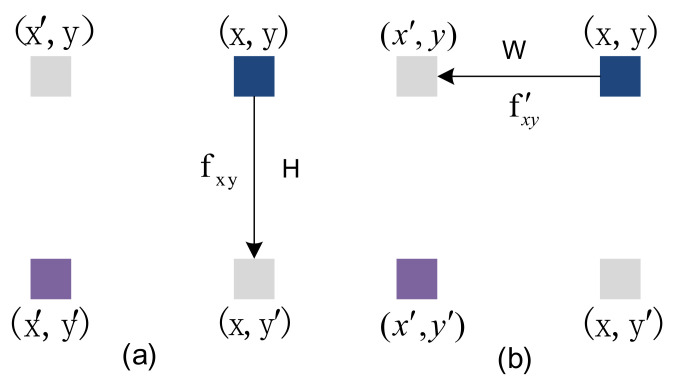
An example of position information propagation, (**a**) represents column-adjacent attention, (**b**) represents row-adjacent attention.

**Figure 5 sensors-22-04477-f005:**
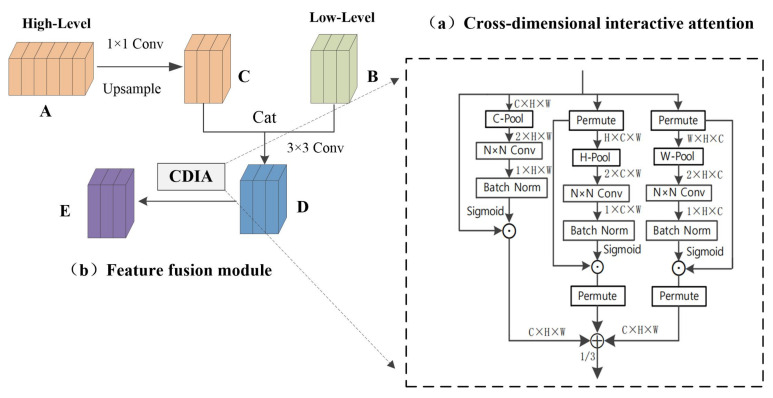
Cross-dimensional interactive attention feature fusion module (CDIA-FFM).

**Figure 6 sensors-22-04477-f006:**
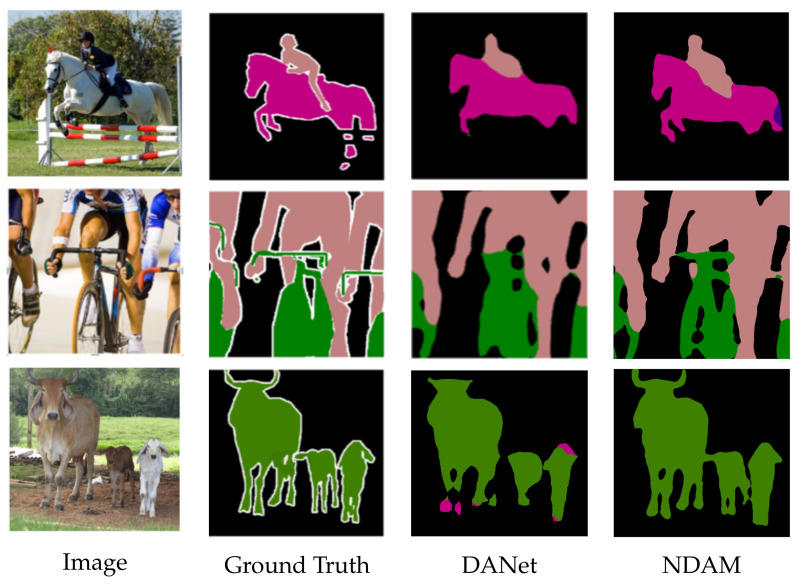
Visual comparison chart of NDAM and DANet.

**Figure 7 sensors-22-04477-f007:**
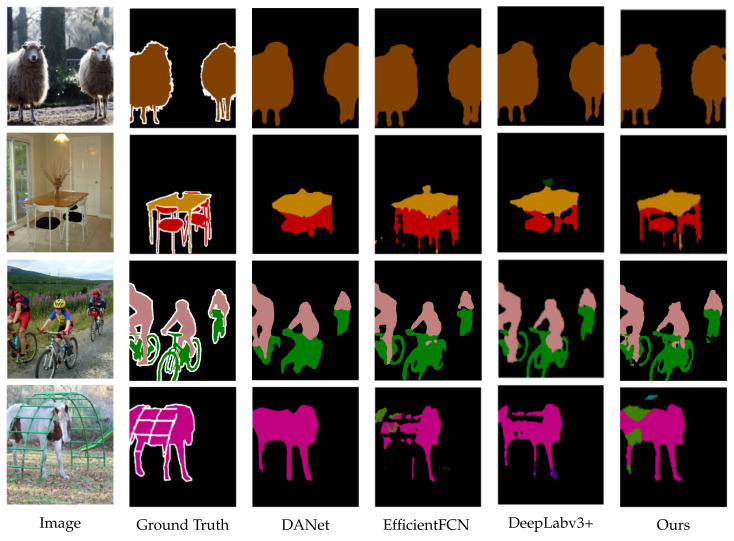
Visual comparison on PASCAL VOC 2012 Dataset.

**Figure 8 sensors-22-04477-f008:**
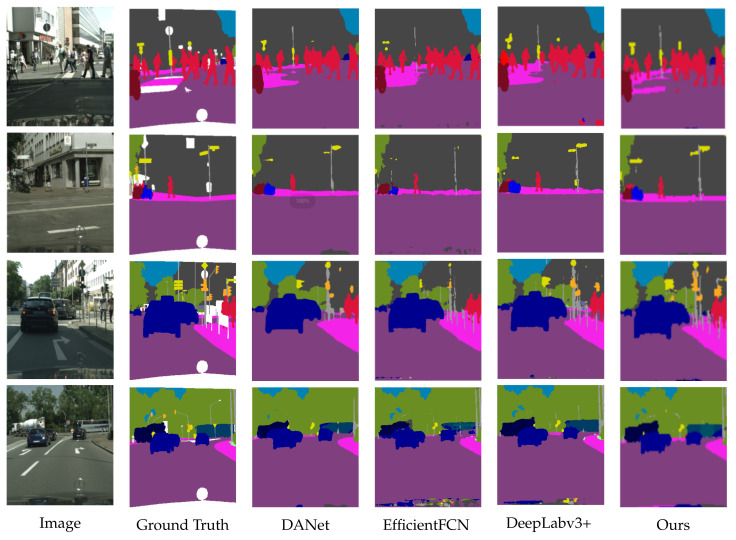
Visual comparison of cityscapes dataset.

**Table 1 sensors-22-04477-t001:** Ablation experiment on PASCAL VOC 2012 dataset, H,W=33 means taking the entire row and the entire column in turn when taking adjacent pixels for similarity calculation. H,W=▵ means that the number of adjacent pixels varies with the position of the pixel when the adjacent pixels are used for similarity calculation, with more in the middle and fewer on the edge.

Method	BaseNet	mIoU (%)
PAM [8]	ResNet101	72.80
APAM $\left(H,W=▵\right)$	ResNet101	73.15
CAM + PAM [8]	ResNet101	73.39
CAM + APAM $\left(H,W=33\right)$	ResNet101	73.13
CAM + APAM $\left(H,W=▵\right)$	ResNet101	73.57

**Table 2 sensors-22-04477-t002:** To evaluate the model complexity of NDAM on the PASCAL VOC 2012 dataset, we use 1 × 3 × 513 × 513 feature maps as input.

Method	BaseNet	Parameters	FLOPs
DANet [8]	ResNet101	66.327 (M)	80.774 (G)
NDAM	ResNet101	65.294 (M)	79.738 (G)

**Table 3 sensors-22-04477-t003:** Ablation experiments on the PASCAL VOC 2012 data set. The parameter *N* represents the size of the convolution kernel in the cross-dimensional interactive attention decoding module. There are three fusion modules in the network.

Method	*N*	mIoU (%)
CDIA-FFM	3, 3, 3	75.10
CDIA-FFM	5, 5, 5	75.34
CDIA-FFM	3, 5, 7	75.50

**Table 4 sensors-22-04477-t004:** Segmentation results on PASCAL VOC 2012 validation set.

Method	BaseNet	mIoU (%)
Dilated FCN	ResNet101	72.40
DeepLabv3+ [9]	ResNet101	75.45
DANet [8]	ResNet101	73.39
OCRNet [17]	ResNet101	74.69
EfficientFCN [33]	ResNet101	73.78
Bisenet v2 [34]	no	73.88
Ours	ResNet101	75.50

**Table 5 sensors-22-04477-t005:** Segmentation results on cityscapes validation set.

Method	BaseNet	mIoU (%)
Dilated FCN	ResNet101	66.61
DeepLabv3+ [9]	ResNet101	71.21
DANet [8]	ResNet101	69.08
OCRNet [17]	ResNet101	68.14
EfficientFCN [33]	ResNet101	68.65
Bisenet v2 [34]	no	68.50
Ours	ResNet101	72.80

**Table 6 sensors-22-04477-t006:** Compared with transformer method.

Method	Backbone	PASCAL VOC 2012 (mIoU%)	Cityscapes (mIoU%)
Segformer	Mit-B2	74.9	78.1
Ours	ResNet101	75.5	72.8

## Data Availability

The data presented in this study are available on request from the corresponding author.

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
