# Peer review of "Multiple-Attention Mechanism Network for Semantic Segmentation"

_sensors, 2022, doi:10.3390/s22124477_

Round 1
Reviewer 1 Report
- The whole paper is a bit confusing, especially the third chapter describing the model is unreasonably structured.
- Some key points in the paper are not explained in detail. For example, in Section 3.1, some key parameters that appear for the first time are not explained for the first time.
- The author needs to further sort out the ideas and re-adjust the structure of the paper.
- In addition, there are some low-level mistakes, such as spelling mistakes, abbreviations are not explained, etc.
Author Response
Response to reviewer #1:
We really appreciate you for your carefulness and conscientiousness. Your suggestions are really valuable and helpful for revising and improving our paper. According to your suggestions, we have made the following revisions on this manuscript:
1.(The whole paper is a bit confusing, especially the third chapter describing the model is unreasonably structured.)
Response:Thank you very much for your advice. We restructured Chapter 3, which includes three sections: 3.1 Network Architecture, 3.2 New dual attention model, 3.3 Cross-dimensional interactive attention feature fusion module. We marked these titles in blue font.
- (Some key points in the paper are not explained in detail. For example, in Section 3.1, some key parameters that appear for the first time are not explained for the first time.)
Response: Thank you for your constructive comments. In Chapter 3, we add some detailed explanations at key points, such as some key parameters K, Q and V; C, H and W, and N, etc. (lines 201-210). In Section 3.2.1 we reworked our expression, and the revisions are marked in blue font.
- (The author needs to further sort out the ideas and re-adjust the structure of the paper.)
Response:Thank you for your positive comments.We readjusted the structure of the paper, mainly revised the third chapter, and rethought the logical relationship of the third chapter. And the redundant introduction to Chapter 3 is placed in the introduction chapter (lines 48-55).
- (In addition, there are some low-level mistakes, such as spelling mistakes, abbreviations are not explained, etc.)
Response:Thank you for the suggestion. We have checked spelling issues and unexplained abbreviations, and made relevant corrections and additions. For example: we have explained NDAM, CDIA-FFM (lines104-109), we have also made a brief introduction to the abbreviations in the Introduction chapter and related work chapters, and cited relevant literature for readers to understand.
Reviewer 2 Report
The authors present a study in the line of semantic segmentation methodologies. The study is exciting, actual and was well conducted, and in general, the presented results are clear and understandable. However, some aspects can be more clarified and adjusted to give the readers a better understanding, particularly the new ones. This can be improved and complemented in the writing of the work.
Please check the following.
Acronyms
All the acronyms need to be checked and fully described. Examples: U-Net, SegNet, DFN, PAN, APAM, NDAM, CDIA-FFM, AFF, Uet, OCRNet, ……. This will better understand the paper and get better interest to new readers in the domain.
Other examples:
Line 8: Authors have in the abstract the indication of (C,H), (C,W), (H,W). What is the meaning of the variables? Is it necessary to provide this information in the abstract? I suggest removing it.
Line 11: What is the meaning of mIoU?
Line 22: FCN must have a description of its meaning.
Line 43: GPU – what is the meaning?
Grammatical Errors
The authors opt for the first person focused writing. This option is correct. However, along the text is identified an excess of the same use. It is just a comment, in the sense of a reflection and eventual attempt to adjust.
Several grammatical errors have been identified that need to be adjusted. A general review is required for the text. For example, Line 4 : Check grammatical errors. Line 133: Please remove “etc.”
Figures
The figures presented are interesting but have an incomplete description, which can be seen as a reduction in the interpretation of the work. There should be an adjustment and the information add-on and the verification that all figures are described in the text.
Example: figure 1 – why the legend doesn’t have the description of (a) and (b)?
Figure 2 is not indicated in the text description.
Author Response
Response to reviewer #2:
Thank you for reviewing our manuscript and offering valuable advice. In accordance with your suggestions, we have made the following revisions to our manuscript:
- (cronyms :All the acronyms need to be checked and fully described. Examples: U-Net, SegNet, DFN, PAN, APAM, NDAM, CDIA-FFM, AFF, Uet, OCRNet, ……. This will better understand the paper and get better interest to new readers in the domain.)
Response:Thank you for your insightful comment. The explanation of APAM is in lines 62 to 67, and we explain NDAM, CDIA-FFM in lines104-109. For acronyms like U-Net, SegNet, DFN, PAN, AFF, OCRNet, etc., we also do Brief introduction, because the introduction chapter and related work chapters should not be too long, so there are relevant citations after the acronym for the reader's understanding.
- (Line 8: Authors have in the abstract the indication of (C,H), (C,W), (H,W). What is the meaning of the variables? Is it necessary to provide this information in the abstract? I suggest removing it.)
Response:Thank you very much for your valuable comments. We have removed relevant content and made relevant changes (lines 4-10).
- (Line 11: What is the meaning of mIoU? Line 22: FCN must have a description of its meaning. Line 43: GPU – what is the meaning?)
Response:We are grateful for the suggestion. The explanation of mIOU is in lines 281 to 285 of the article, and we marked it with red font; we added the meaning of FCN in lines 23-28; and we changed the expression of GPU in line 288, GPU is a general term for experimental equipment.
- (Grammatical Errors:The authors opt for the first person focused writing. This option is correct. However, along the text is identified an excess of the same use. It is just a comment, in the sense of a reflection and eventual attempt to adjust. Several grammatical errors have been identified that need to be adjusted. A general review is required for the text. For example, Line 4 : Check grammatical errors. Line 133: Please remove “etc.”)
Response:Thank you for the suggestion. We have checked for relevant grammatical errors and corrected them.
- (Figures:The figures presented are interesting but have an incomplete description, which can be seen as a reduction in the interpretation of the work. There should be an adjustment and the information add-on and the verification that all figures are described in the text.
Example: figure 1 – why the legend doesn’t have the description of (a) and (b)?
Figure 2 is not indicated in the text description.)
Response:Thank you for underlining this deficiency. We have added some descriptions of Figure 1 on lines 69-85, and we have a text description of Figure 2 on lines 181-187 of the article.
Round 2
Reviewer 1 Report
The overall logic is a little confusing, and it doesn't appear that anything has changed since the prior version. Figure 2 is the overall structure diagram, for example, while the model diagram in Figure 3 (and later) does not align well with the overall structure diagram, causing the viewer to be perplexed.
The presentation might use some work. Many parts, including the Abstract, are not smooth.
Author Response
We really appreciate you for your carefulness and conscientiousness. Your suggestions are really valuable and helpful for revising and improving our paper. According to your suggestions, we have made the following revisions on this manuscript:
1.(The overall logic is a little confusing, and it doesn't appear that anything has changed since the prior version.)
Response:Thank you very much for your advice. We have revised the content of Chapter 3 again, and the revised content is marked in blue font. The overall logic of our proposed MANet semantic segmentation network is introduced in 3.1 network architecture, figure 2 can also be a good understanding of our overall framework. The new dual attention model (NDAM) and Cross-dimensional interactive attention feature fusion module (CDIA-FFM) are described in sections 3.2 and 3.3, respectively, and the model structure diagrams are shown in Figure 3 and Figure 5.
- (Figure 2 is the overall structure diagram, for example, while the model diagram in Figure 3 (and later) does not align well with the overall structure diagram, causing the viewer to be perplexed.)
Response: Thank you for your constructive comments. We have reorganized the model diagram of Chapter 3 to make the whole structure more logical. In the previous version, there was an error in the title of Figure 4, which we have corrected. As shown in Figure 5, we have refined the structure diagram of Section 3.3.
- (The presentation might use some work. Many parts, including the Abstract, are not smooth.)
Response:Thank you for your positive comments. We rethought and polished the abstract of the paper, and corrected grammar and presentation in the paper.